# Physical Model Tests of Concrete Buttress Dams with Failure Imposed by Hydrostatic Water Pressure

**Jonas Enzell** [1,*] , **Erik Nordström** [1] , **Andreas Sjölander** [1] , **Anders Ansell** [1] and **Richard Malm** [2]

1 Department of Civil and Architectural Engineering, KTH Royal Institute of Technology, Brinellvägen 23, 100 44 Stockholm, Sweden
2 Department for Weapons, Protection and Security, FOI Swedish Defence Research Agency, Olof Arrhenius väg 31, 137 94 Norra Sorunda, Sweden
* Correspondence: jonas.enzell@byv.kth.se

**Abstract:** Although the failure of a concrete dam is a complex and highly dynamic process, the current safety assessments of concrete gravity and buttress dams rely on a simplified 2D stability analysis, which neglects the load redistribution due to 3D monolith interactions and the valley shape. In addition, the estimation of breach parameters in concrete dams is based on assumptions rather than analyses, and better prediction methods are needed. Model tests have been conducted to increase the understanding of the failure behavior of concrete dams. A scale model buttress dam, with a scale of 1:15, consisting of 5 monoliths that were 1.2 m in height and 4 m in width, was constructed and loaded to failure using water pressure. The model dam had detachable abutment supports and shear keys to permit variations in the 3D behavior. The results showed that the shear transfer was large between the monoliths and that the failure of a single dam monolith is unlikely. A greater lateral restraint gives not only a higher failure load but also a better indication of impending failure. These findings suggest that the entire dam, including its boundary conditions, should be considered during a stability assessment. The results also suggest that the common assumption in dam safety codes that a single monolith fails during flooding analysis is not conservative. The dataset obtained provides a foundation for the future development of dam-monitoring alarm limits and for predictive models of dam-breaching processes.

**Keywords:** concrete dams; buttress dams; physical model tests; dam failures; stability assessment

## 1. Introduction

Few types of failure pose as high a potential for catastrophic outcomes as a large dam failure, which may lead to the uncontrolled release of a large amount of water and thus to enormous property damage and loss of life. Dam failures have been relatively rare compared to the failures of other engineering structures, with most documented instances occurring in the early 20th century [1]. There are few well-documented dam failures that could serve as references for, e.g., the verification of numerical simulations [2].

A dam failure is a complex and highly dynamic process influenced by many factors, including irregular geometries, nonlinear material behavior, the dynamic effects of the water release, dam structure movement, and rock and soil erosion, but safety assessments of concrete gravity and buttress dams are based on simplified analytical stability methods [3,4] performed in 2D for one monolith at a time, typically with a simplified rock surface. This means that 3D effects, such as the influence of the valley shape and the interaction between monoliths, are neglected [5,6]. The present study explores the three-dimensional behavior of dams and the impact of water on the structure during failure using physical model tests.

In flooding simulations used for emergency preparedness planning and risk evaluations, the breach size and development time are important as they directly influence the dam-breach discharge hydrograph. Methods exist for assessing the development of failures

in embankment dams [7–9], but the breach parameters for concrete dams are usually based not on analyses but on simplified assumptions. The breach size assumption varies widely between national regulations, between one or two monoliths [9], a minimum of three monoliths [10], or the entire dam [11]. In the guidelines, the breach is considered to occur instantly, although ICOLD [9] mentions that gravity dams are assumed to have a "short but not instantaneous" failure time, whereas buttress dams are usually assumed to fail instantly. Veale and Davison [2] attempted to perform a statistical estimation of the breach geometry of concrete gravity dams but concluded that the scatter in the available dataset was too large.

Finite element (FE) modeling is often used for complex structural analyses, but a significant drawback of numerical methods is the limited possibilities for validation. Validating the results of numerical models is difficult when limited data are available from previous cases. Physical model tests offer an approach to validate numerical simulations, and the current work aims to provide a large dataset for validation. Physical model tests have previously been used to validate the results of numerical analyses of concrete dams [12–14]. It is difficult to fully describe the dam breach development for a concrete dam, including the dynamic effects of the uncontrolled release of water, using numerical analyses. Nonetheless, it would be beneficial to validate the results against well-documented physical model tests. A preliminary step could be to examine the 3D behavior of the dam and the impact of the water on the structure during failure using physical model tests.

Monitoring is an essential tool for understanding dam behavior and preventing failure. The monitoring system is often provided with alarm limits to allow the dam owner to react, take action, and prevent failures, but there is no well-defined way to determine the alarm limits, and it is often left to the judgment of the engineers [15]. A distinction can be made between the displacements before and after the failure. A pre-failure displacement will probably precede a dam failure, but, at some point, the failure will develop out of control, and a breach will occur. This study intends to enhance the understanding of failure behavior to provide a basis for determining alarm limits in the future. If a pre-failure displacement occurs and is detected, there is a better chance of preventing a total failure. An increased understanding of pre-failure displacement would make it possible to determine alarm limits and increase dam safety.

The present work seeks to increase the understanding of dam failure events to enable the development of better analysis methods for determining dam safety. With this aim, physical model tests of a concrete buttress dam have been conducted to examine the 3D behavior of the dam, to determine the existence and extent of pre-failure displacement, and to estimate the breach size and development time. A 1:15 scale model concrete buttress dam was constructed and loaded to failure using hydrostatic pressure. The model had an idealized geometry representing a general case. It did not account for phenomena such as cohesion or seepage in the dam body, which vary significantly between individual dams. Modular shear keys and side supports were used to investigate the interaction between monoliths and the influence of the boundary conditions. The results showed that the influence of the shear key and boundary conditions is substantial, suggesting that 3D behavior should be considered in the stability analysis of a concrete buttress dam. The results of these model tests comprise a large dataset that can be used to validate and calibrate future finite element (FE) models.

This study addresses a gap in the understanding of the failure behavior and 3D effects on concrete dams through the use of a physical model test using a model (1:15) concrete dam loaded using water pressure. Previous research has primarily focused on the behavior of individual monoliths and the ultimate limit state. The findings challenge prevailing assumptions in dam safety codes. This study provides data that could validate and calibrate existing numerical models, and the results also emphasize that it is important to consider 3D effects when making dam safety assessments. The results also offer a previously unattainable insight into breach development and formation times. The analysis

of pre-failure displacements also introduces a novel perspective regarding early indicators of imminent dam failure.

## 2. Background

### 2.1. Stability Criteria

Concrete gravity and buttress dams are usually assumed to fail along a sliding plane, acting as a rigid body [3,4]. The stability criteria are commonly assessed for two different failure modes, sliding and overturning failure. The limits for stability are given in national or regional codes and regulations, e.g., FERC [3]. If interface cohesion is neglected, the sliding stability can be defined by the following equation:

$$s_{fs} = \tan \delta_g \frac{\sum V}{\sum H} \tag{1}$$

where $\delta_g$ is the friction angle in the sliding plane, $\sum V$ is the normal force, and $\sum H$ is the tangential force along the sliding plane. The safety factor for overturning is defined as the ratio between the stabilizing and destabilizing moments:

$$s_{fo} = \frac{M_s}{M_d} \tag{2}$$

where $M_s$ is the moment from the stabilizing forces and $M_d$ is the moment from the destabilizing forces. The overturning failure mode is used in many national guidelines and regulations, e.g., RIDAS [16] and NVE [17], but FERC [3] and USACE [4] do not recommend using overturning stability. The cohesion between the concrete dam and the rock foundation is often utilized in dam safety calculations [3,4], but, due to the uncertainty in determining cohesion, larger safety factors are required. The strength of the cohesive bond is one of the primary sources of uncertainty in a stability analysis. It is difficult to measure, and a weak spot in the cohesion reduces the shear strength. It is, therefore, often neglected in the analysis [3,18].

For complex geometries or nonlinear material behavior, a stability analysis can be carried out using numerical analysis or physical model tests [19,20]. When using models, the resistance from the structure cannot be directly determined, but failure is instead simulated using a pushover analysis, and the safety factor is calculated as the ratio of the failure load to the design load:

$$S_n = \frac{failure\ load}{design\ load} = \frac{(1 + \lambda)P}{P} = 1 + \lambda, \quad \lambda > 0 \tag{3}$$

### 2.2. Failure of Gravity and Buttress Dams

Photos of a typical Swedish buttress dam are presented in Figure 1. To permit thermal expansion and contraction, gravity and buttress dams are usually divided into distinct monoliths, the expansion joints being sealed using water stops. Shear keys are a common feature, but they are not universally applied. A stability analysis is performed for one monolith at a time without considering 3D effects. Three-dimensional analyses are commonly performed on arch dams and gravity dams with curved axes, but 3D effects have been observed in gravity dams with straight axes [6,21]. Analytical methods have also been developed for assessing the 3D effects on concrete gravity dams [5,22,23]. Stability analyses of entire gravity dams, including their 3D behavior, have been performed using FE simulations [24], but, to the authors' knowledge, the 3D behavior of concrete buttress dams has not been examined.

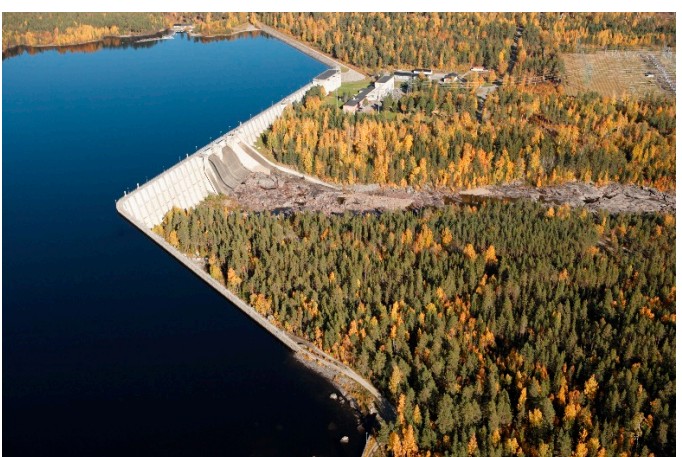 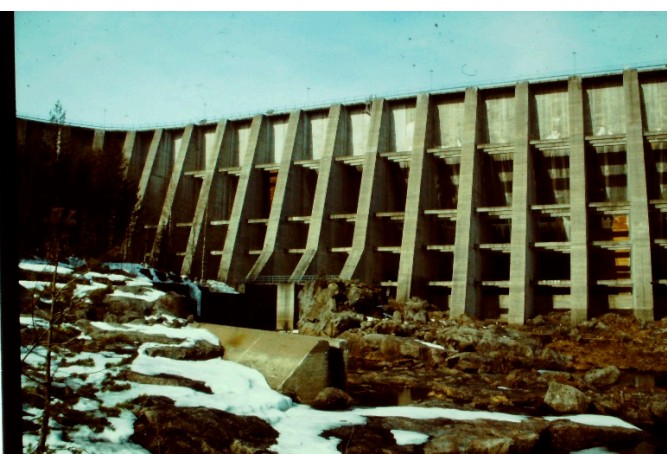

**Figure 1.** Photos of a Swedish concrete buttress dam connected to an embankment dam on one end and the intake building on the other end. At the center of the dam, three spillways are located. Photos: Uniper.

The shear transfer between the monoliths will influence the failure mode of a concrete gravity or buttress dam, but the exact amount of shear force transferred between the monoliths is uncertain, due to the inherent complexity of these structures. The presence of shear keys increases the shear transfer, and USBR [23] notes that the monoliths rotate around a vertical axis during failure. This rotation leads to interlocking between the monoliths and to pressure in the joints. If the dam is restrained in the cross-valley direction, the interlocking can become considerable, leading to an increased transfer of shear forces. This restraint depends on the boundary condition at each end of the dam, where a narrow valley leads to more restraint than a wide valley. Most dams have inlet structures and spillways that connect to the main dam body, and these also influence the overall stability of the structure. These are often massive concrete structures, much heavier than a regular buttress monolith, and they are likely to produce a stiff boundary. Connections to embankment dams are also common, but these connections likely provide less stiffness than concrete structures.

The extent of lateral restraint on the dam is determined by the stiffness of the boundary conditions and the interaction between the monoliths. The lateral restraint creates a 3D effect, which cannot be taken into account in a regular 2D stability analysis. The degree of restraint probably affects the failure process and the breach size. A comprehensive understanding of these phenomena is crucial if the failure process is to be accurately predicted.

*2.3. Physical Model Tests and Scaling Laws*

When the ultimate strength of a structure is determined using a physical model, a pushover test is often used, where the load is increased until failure occurs. In a pushover test, hydraulic jacks are commonly used to apply the load to the structure, but the dynamic effect of the water in the reservoir is neglected. This method is sufficient to determine the ultimate strength of the structure, but it does not accurately simulate the post-peak behavior. For embankment dams and fuse plugs, physical model tests have been performed where the dam has been overtopped and the breach development examined [9,25]. Spillway discharge is also a well-researched area [26,27]. In physical model tests on concrete dams, researchers have used water when assessing the seismic stability of hydraulic structures on shake tables [28,29], as well as in underwater blast loading [30]. No examples have been found where hydraulic structures have been loaded to failure using water pressure. This has, however, been performed within the field of nuclear engineering, where a reactor containment structure was loaded to failure using internal water pressure [31].

Similitude, achieved through proper scaling derived from dimensional analysis, ensures accurate representation in scale models [20,32]. This concept is based on Buckingham's Pi theorem, which establishes relationships between scaled properties [33]. The

similitude requirements for linear–elastic, time-dependent models are given in Table 1, where "linear-elastic" indicates a material model where stress is proportional to strain, and "time-dependent" indicates a model where the response varies with time.

**Table 1.** Scale factors required for linear–elastic, time-dependent physical models, according to Harris and Sabnis [32].

| Quantity | Dimension | Scale Factor |
| --- | --- | --- |
| Length | L | $S_l$ |
| Time | T | $S_l^{1/2}$ |
| Stiffness | $FL^{-2}$ | $S_E$ |
| Weight | $FL^{-3}$ | $S_E/S_l$ |
| Friction | - | 1 |

## 3. Physical Model Tests

In this project, a series of physical model tests were performed, where a dam failure was simulated, including the dynamic effect from the water. The model dam was designed to fail in a sliding mode, with failure being induced by increasing the water level in the model reservoir until a breach occurred. The model consisted of five concrete buttress monoliths, as shown in Figure 2. The monoliths were placed on a concrete slab, representing the foundation. The restraint was varied by changing the boundary conditions at the abutments and the interaction between the monoliths.

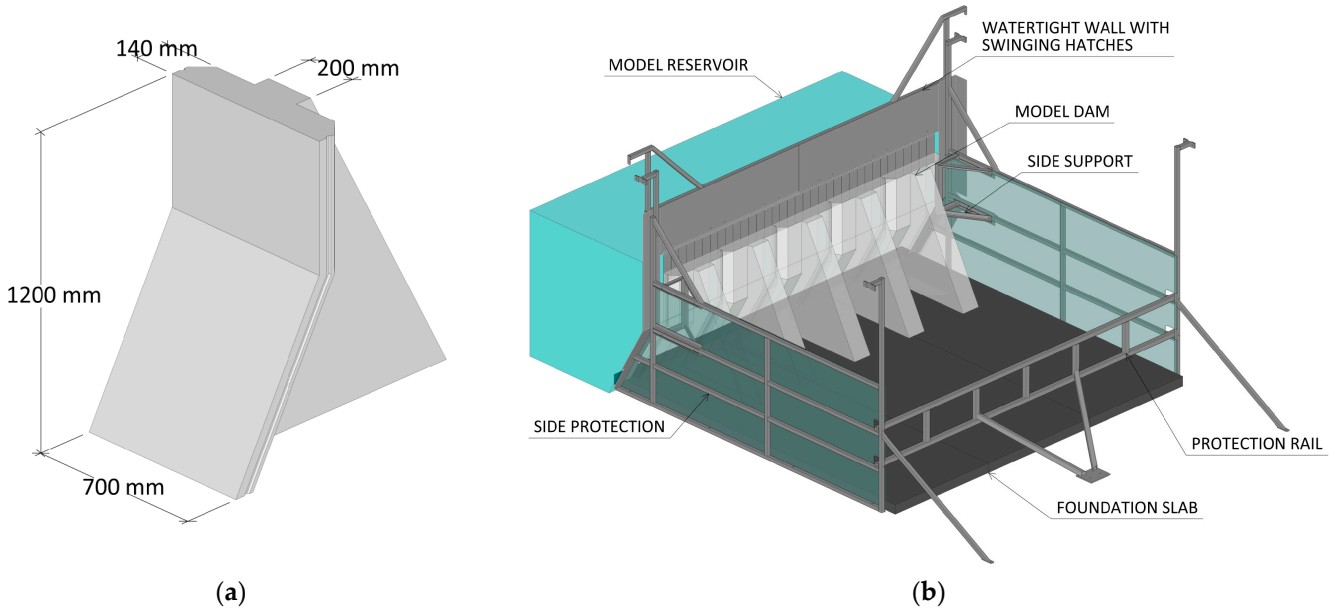

**Figure 2.** The physical model: (**a**) a model monolith and (**b**) the full setup with steel frame and protection rails. Measurements in mm.

The tests were performed in a 25 m long and 4 m wide chute with a maximum depth of 2 m, as shown in Figure 3. The chute is at the Älvkarleby Laboratory, owned and operated by the Swedish power company Vattenfall. The reservoir had a maximum volume of 101 m³, and at the crest level of the model dam, the reservoir contained 66 m³ water. A watertight steel sheet wall was erected upstream of the model dam to allow the water level to rise above the crest of the dam. The sides of the model area were enclosed with transparent plexiglass mounted on a steel frame to protect the chute. The downstream movement of the monoliths was limited by a steel beam at the end of the foundation slab; see Figure 2.

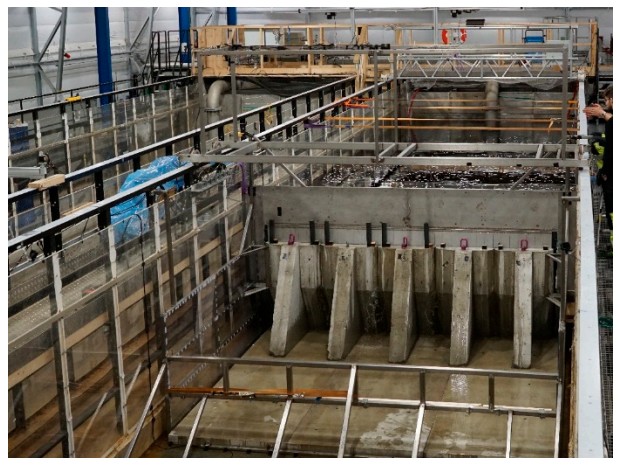
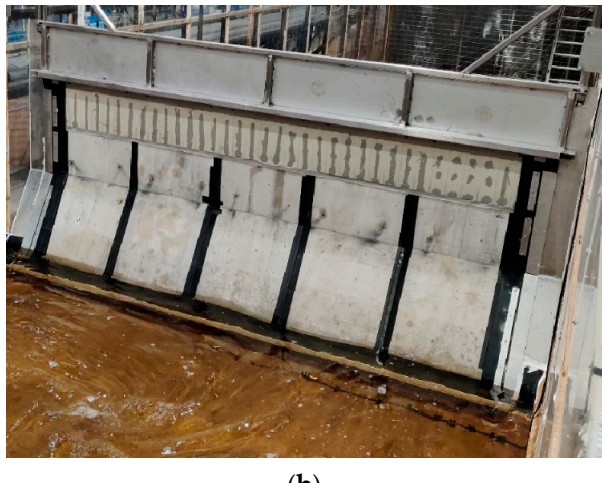

(**a**)　　　　　　　　　　　　　　　　　　　　　　　　　　(**b**)

**Figure 3.** Photographs of the chute and the model showing (**a**) the downstream and (**b**) the upstream views. The rubber sheets used to seal the model are visible in image (**b**).

### 3.1. Geometry

A prototype of a concrete buttress dam representative of several Swedish buttress dams in size, shape, and slenderness was constructed. The model monolith was created by scaling down the prototype by a factor of 1:15, resulting in a monolith that was 1.2 m high and 0.7 m wide; see Figure 2a. The front plate and the buttress wall had thicknesses of 140 mm and 200 mm, respectively. These thicknesses were slightly greater than desired but necessary to ensure a sufficient weight for stability reasons. Each monolith weighed ca. 570 kg. The concrete slab representing the rock foundation was 100 mm thick and cast with plywood formwork to create a smooth surface. The foundation slab measured 3.7 m in both length and width and was divided into three sections, with joints parallel to the stream direction.

Rubber seals were installed at all joints on the upstream side of the model dam; see Figure 3b. The seals did not make the joints perfectly watertight, but the model was sufficiently watertight to enable the model reservoir to be filled. Water leakage started to occur before any displacement of the monoliths, and after some initial displacements, the leakage increased considerably. The increased leakage was nevertheless small enough to allow the model reservoir to be filled.

### 3.2. Boundary Conditions and Test Series

A primary objective of this study was to investigate the 3D behavior of the model dam and to examine the effects of the boundary conditions and the interaction between the monoliths. To achieve this, the model setup was varied using detachable shear keys and abutment supports. The monoliths were cast with keyways on both sides of the front plate, and the shear keys were produced from ABS plastic and screwed in place; see Figure 4a. The shear keys protruded 10 mm. This design was taken from a Swedish buttress dam. Boundary conditions at the abutments were set by adding steel side supports behind the vertical part of the outer monoliths; see Figure 4b. These side supports were removable, allowing the boundary condition to be applied on one or both sides or to be removed entirely. These boundary conditions were rigid, preventing any downstream displacement. The edge monoliths were allowed to move sideways and to rotate, so that the dam failed at reasonable water levels.

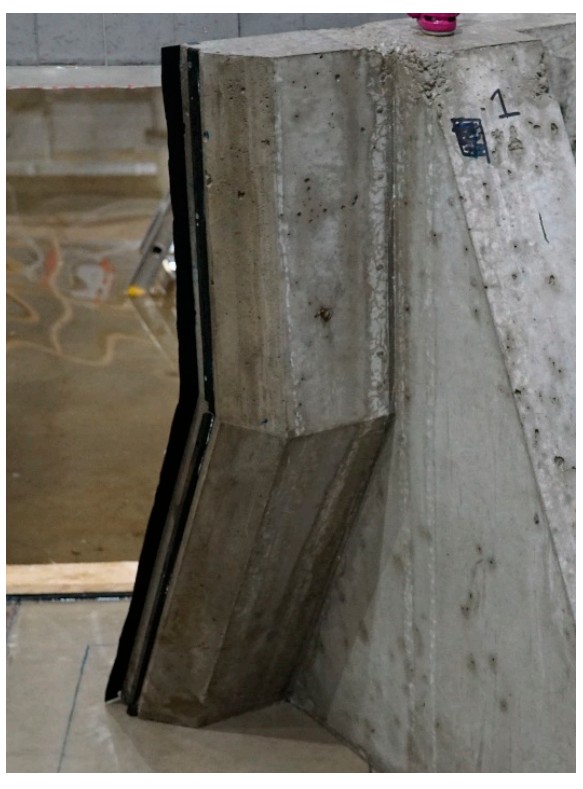

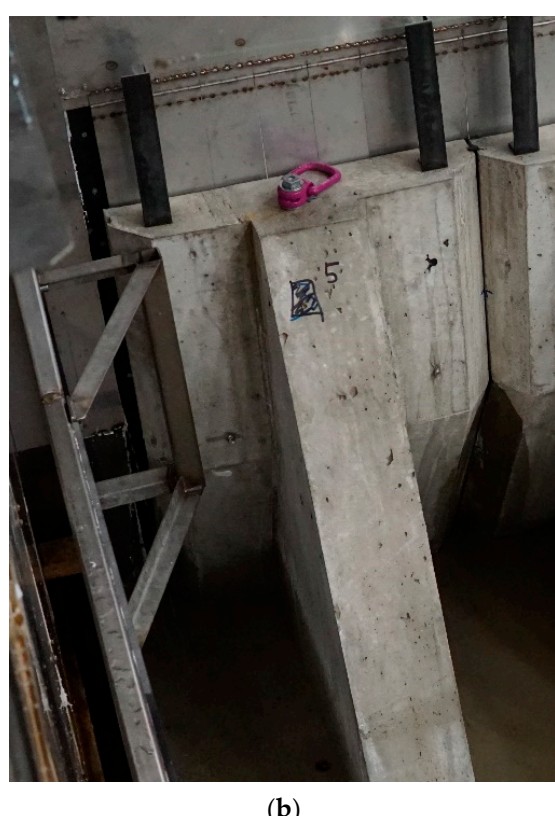

(**a**)                                                                                          (**b**)

**Figure 4.** Details of the model: (**a**) the shear keys and (**b**) the side supports behind the outer monoliths.

The detachable shear keys and side supports made it possible to vary the setup and to run different test series. The test series are presented in Table 2, grouped according to the degree of lateral restraint as described in the background section. The test with the failure of a single monolith (SM) was taken as the reference case. This case was in agreement with the sliding stability criteria of Equation (1). Cases where the entire model dam failed are designated full dam (FD). The case without shear keys and without side supports (FD) and the case with shear keys but no side supports (SK) are grouped together as having a low lateral restraint. These two cases simulate a long buttress dam in a wide valley without rigid boundaries or large intake structures.

**Table 2.** Test series and boundary conditions.

| Test Series | Acronym | Description | Side Supports | Shear Keys | Number of Tests |
|---|---|---|---|---|---|
| Single monolith | SM | Failure of Monolith 3 | yes | 1–2 & 4–5 | 3 |
| Low restraint (LR) | FD | No supports or shear keys | no | no | 3 |
| | SK | Shear keys | no | yes | 6 |
| Moderate restraint (MR) | SP | Side supports, no shear keys | yes | no | 6 |
| | SP(OT) | Same as SP but OT allowed | yes | no | 3 |
| High restraint (HR) | SP+SK | Side supports and shear keys | yes | yes | 6 |

Side supports were introduced to increase the degree of lateral restraint at the boundaries. Test series were performed without shear keys (SP) (achieving a moderate degree of restraint) and with shear keys (SP+SK) (achieving a high degree of restraint. These conditions represent a shorter buttress dam situated in a narrow valley or close to a massive intake structure. In the overtopping test series (SP(OT)), the watertight wall was removed, and the side supports were used but no shear keys. This case was included to simulate the effect of overtopping and to assess the influence of the watertight wall. Rock and soil erosion downstream from the dam toe is the main risk of overtopping, but it was not

possible to recreate this in these tests. To control for the effect of model wear, several test series were repeated twice. When relevant, the test number is given after the acronym denoting the test series.

In total, 42 tests were performed, but the results from 17 of the tests were discarded. That includes two tests of the pumps and watertightness of the model, two failed attempts for the overtopping tests, two failed overturning tests, and two failed tests due to wear on the model with high restraint (see Figure 5). The first three SM tests failed because the central monolith was not physically separated from the adjacent ones, and the results were omitted. Finally, six tests were performed with a single support. These did not result in the restraint assumed in the initial design of the tests. They instead resembled the tests with low restraint. They were, therefore, not considered in this study. However, the tests with one support were used to assess the applicability of digital image correlation (DIC) in the presented model tests [34]. This is presented further in Section 3.5.

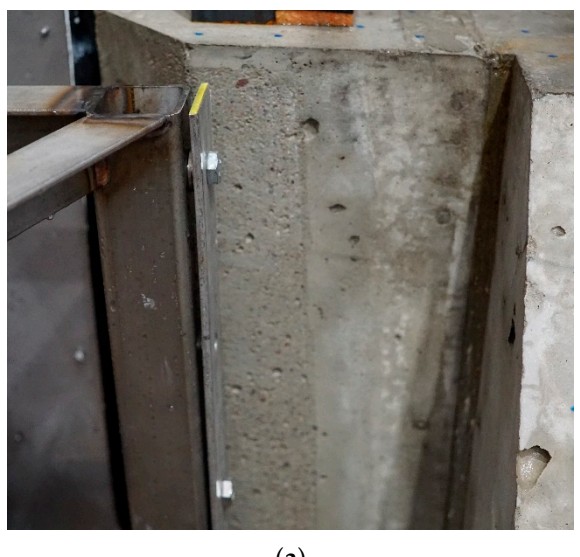 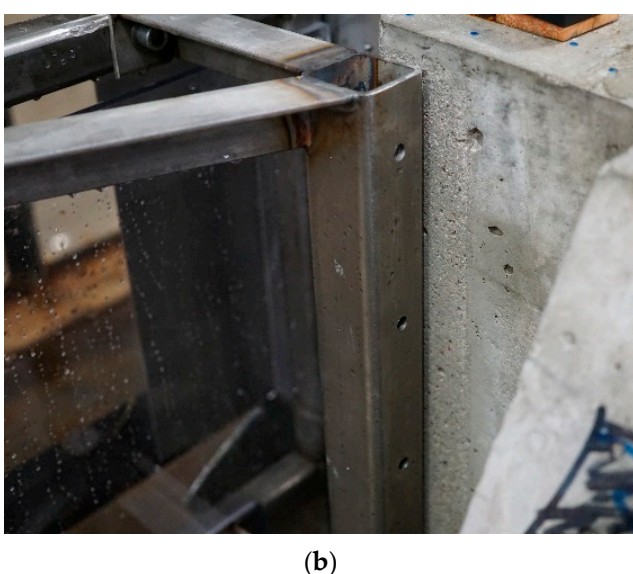

| (**a**) | (**b**) |

**Figure 5.** The right-hand side support (**a**) with the extension and (**b**) without the extension but with a roughened concrete surface.

The support on the right-hand side was a square steel pipe with rounded corners, and it initially had an extension making it 10 mm wider. The extension was made from a flat steel bar screwed to the support; see Figure 5. The edge of the extension was relatively sharp, and it therefore wore down the contact surface of the monolith on the right-hand side. When the second round of tests was performed, the rough concrete surface led to higher friction and the lateral restraint became so high that the model did not fail, even at the maximum water level of the chute. The extension was therefore removed. This reduced the friction at the support, but made it difficult to compare the results from the first and second rounds of tests. This affected only the tests using side supports (SP, SP+SK, and OT).

### 3.3. Scaling

The chute's size restricted the scale of the model, necessitating a balance between the height of the model and the number of monoliths. A 1:15 scale was used, resulting in a relatively high model with five monoliths, which is the minimum number required to achieve a variation between the different test series. The gravity loads were scaled by 1:1. If a material failure is simulated, the material strength must be downscaled by 1:15 to fulfill the similitude requirements, as outlined in Table 1. The model, however, failed in a sliding mode. The decisive material property was therefore the friction at the interface between the dam and its foundation, which is unitless and scales 1:1, and the monoliths could thus be made from regular strength concrete, eliminating the need for materials with scaled-down

strength. This results in a condition similar to the rigid body assumption adopted in the stability criteria in Equations (1) and (2). The use of regular-strength concrete eliminated problems associated with the use of downscaled concrete, such as matching the fracture energy with the compressive strength. This also enabled the same model monoliths to be used in all the test series.

As discussed by Heller [35] and Yang et al. [36], scale effects arise mainly in hydraulics phenomena. This study investigates pressure-induced structural behaviors, with little focus on hydraulic parameters, such as aeration. The structural results are subjected to minimal hydraulic scale effects.

When the elastic deflection caused by the hydrostatic pressure is considered, it is important to note that the crest deflections caused by the elastic response are small compared to the initial sliding deflections. The elastic deflection is not large enough to create an interlocking effect between the monoliths, which would add significant stiffness to the system, and for the purposes of this model, the contribution of the elastic deflection to the overall system behavior is therefore negligible.

### 3.4. Material and Friction

The model dam and the foundation slab were cast from concrete of strength class C30/37, as defined in *Eurocode 2* [37]. The monoliths and the foundation slab were cast using plywood formwork, giving smooth surfaces at all the interfaces. Cohesion was not considered because of difficulties associated with determining it at the dam–foundation interface and the problems arising from an uneven distribution. During the casting of the monoliths, the formwork became somewhat deformed, so the joints between the monoliths were fitted using a concrete grinder. Since the formwork did not deform during the casting of the foundation slab, some joints protruded in the assembly due to difficulties in fitting the separate partitions together. Surface irregularities were removed by local grinding. The ground surface was smooth and similar to the cast surfaces. Friction tests determined that the effect of grinding on the friction was minor, reducing the friction angle by approximately 4%; see Figure 6.

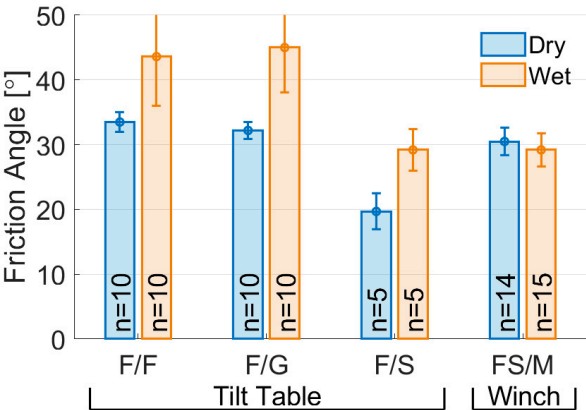

**Figure 6.** Average friction angles with standard deviations. F denotes the form surface, G the ground surface, S the steel bar, FS the foundation slab, and M the monolith.

The friction at the interface between the model monoliths and the concrete slab was evaluated using the tilt table method proposed by Alejano et al. [38], but the tests were performed with a manually operated tilt table, so a fixed tilt rate could not be used. Test blocks for the friction tests, measuring $150 \times 150$ mm$^2$ and sawn to a height of 35 mm, were cast from the same concrete batch as the monoliths and the foundation slab. Since there was some leakage in the model, the interface between the monoliths and the foundation slab became saturated with water. For this reason, the friction tests were performed with both dry and wet blocks. The average friction from the tests is presented in Figure 6.

The mean friction of the wet samples was difficult to assess because of the large scatter, but the friction was greater with wet than with dry samples, likely because of capillary suction. The friction between the concrete and a square stainless-steel pipe was also determined as a measure of the friction between the monoliths and the side supports.

After all model tests were finished, the friction between the model dam and the foundation was estimated by pulling the monoliths downstream using a winch. The friction angle was calculated from the weight of the monolith, $V$, and the force, $H$, required to pull the monoliths downstream:

$$\delta = \tan^{-1}\frac{H}{V} \tag{4}$$

Each monolith was tested three times under both dry and wet conditions, and the results are presented in Figure 6. The winch tests gave the better estimation of the wet friction and of the wear on the model.

The tilt table tests probably overestimate the wet friction due to the small surface area and the perfect fit between the cubes. The model monoliths had a larger interaction area, and the fit was imperfect, which reduces the influence of the capillary forces. Unfortunately, the friction was not tested before the test series was started, so the exact effect of wear could not be determined. Since the wet friction angle was 95% of the dry friction angle, the original wet friction angle, assuming the same relationship, would have been $33.5 \times 0.95 = 31.8$ degrees, which corresponds to a friction coefficient of 0.63.

### 3.5. Instrumentation

During the tests, the hydrostatic pressures in the reservoir and in the runoff area were monitored, as well as the displacement of the model dam. A combination of analog instruments, cameras, and video cameras was used for the monitoring. The hydrostatic pressure was monitored using analog submersible piezometers (PZs) at four points upstream of the model dam and at six points downstream in the runoff area, as shown in Figure 7. STS ATM.ECO/N sensors were used upstream, and STS MTM/N10 sensors were used downstream. The displacement of the model dam was monitored using six potentiometer wire gauges (WGs), mounted on the monoliths at the chute's right-hand side and connected to the edges of the front plate at the crest level; see Figure 7. The piezometers and the wire gauges were synchronized and logged data at a frequency of 10 Hz to logger unit Campbell CR1000.

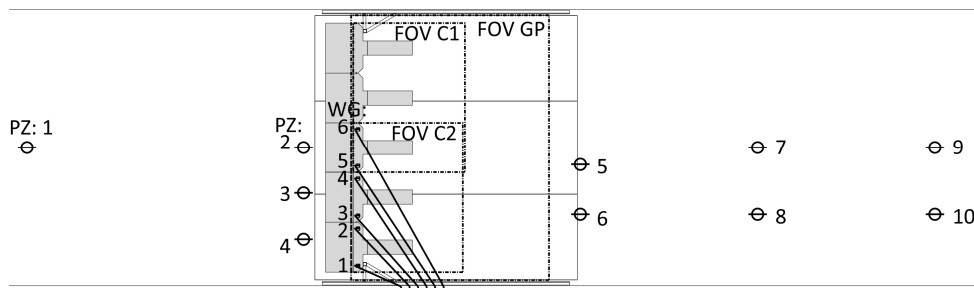

**Figure 7.** The instrumentation placement, including wire gauges (WGs), piezometers (PZs), and the field of view (FOV) of the top view cameras and video cameras. The numbers indicate the sensor number.

Video cameras (6 GoPro Hero 5 and 3 GoPro Hero 9) were installed at nine points around the dam, to overview the model tests, filming with a resolution of 2.7 k with a frame rate of 60 fps. Three cameras (2 Sony α6000, 24 MP, and 1 GoPro Hero 9, 24 MP) and one video camera (GoPro Hero 9, 2.7 k, 60 fps) were installed above the dam to measure the downstream displacement in 2D. The cameras, set up with a time-lapse function, took photographs at a frequency of 0.5 fps. The video camera filmed with a frame rate of 60 fps.

The wire gauges had a measurement range of 1.2 m. However, the model area allowed for a displacement of 3 m. Therefore, the wire gauges were primarily used to monitor the initial displacements, while the displacement during and after the failure was monitored using digital image correlation (DIC) with the py2DIC software [39]. The applicability of DIC to these model tests was evaluated in a master's thesis, where it was concluded that the displacements could be traced with good precision [34]. By using DIC, the displacements could be recreated at different points along the model dam in the post-processing. Using DIC also allowed for the displacements to be extracted in 2D, while the wire gauges can only capture the displacement in 1D. The displacements from the DIC were calibrated with the results from the wire gauges. Since the wire gauges were synchronized against the piezometers, the signal from the DIC could also be synchronized in the post-processing. The photographs from the cameras with 0.5 fps were used to capture the pre-failure behavior, and the frames from the video camera with 60 fps were used to capture the failure of the dam. The field of view (FOV) of the top-mounted cameras is shown in Figure 7. When the camera-mounting rig covered the point of interest in the video, quadratic interpolation was used to obtain the DIC displacement histories. A random speckled pattern is often used in photogrammetric measurements. The surface pattern of the concrete was initially deemed to be sufficient, but a speckled pattern or discrete markers might have enhanced the pattern recognition capability of the DIC software.

### 3.6. Limitations

The model test primarily focused on examining how 3D variables affect the load-bearing capacity of a concrete buttress dam. The breach size, the breach development time, and their implications for dam monitoring were also studied. The load-bearing capacity of a dam is assumed to be dependent on the boundary conditions from adjacent structures and abutments, the foundation properties, and the site topography. In the model tests, the geometry and boundary conditions were, however, simplified towards an idealized case to enhance their general applicability. The model tests therefore lack specificity, so that the results cannot be directly applied to assess the safety of individual dams. As previously mentioned, the model had a limited displacement, but it was nevertheless deemed to be sufficiently large to give accurate results.

The strength of the dam–foundation interface was lower than expected in a real dam. In the model, the friction and surface roughness were lower than what might be expected in the field, and the cohesion at the interface was not considered. Despite these simplifications, the model still provided valuable insights. Importantly, these conditions made possible the induction of failure under a load near the crest level.

Numerous tests were performed, leading to observable wear on the monoliths. This wear is discussed in the results section, but it was difficult to precisely determine its impact on the failure load.

Only one load case was considered: a rapid increase in the reservoir level, which led to a sliding failure along the dam–foundation interface. Failures along fracture planes in the rock and concrete or alternative loading schemes could be considered, but they are outside the scope of this investigation. Cracking of the concrete could affect the behavior of a real dam, but due to the rigid body conditions arising from the use of normal-strength concrete, this effect could not be studied in the model. This will be investigated in future studies. Seepage through the dam body and rock foundation was not simulated, but although the seepage affects the uplift pressure in joints and fractures, it is not expected to affect the failure mode significantly. It was thus considered reasonable to neglect this factor in the model tests.

## 4. Results

Each test series was initially conducted with three tests per series, and some series were repeated, resulting in a total of six tests for these series. Variations in pre-failure and post-breach behavior in the test series are presented in the following sections. Whether or

not the different test series give statistically different results has been tested with Welch's *t*-tests, with $p < 0.05$ as the criterion. The predicted failure load with Equation (1) was compared to the test series using a two-sided Student's *t*-test, with $p < 0.05$ as the criterion. The failure load is expressed in mm of hydrostatic pressure at the base of the dam, which correlates to the water level in the reservoir.

### 4.1. Comparison Based on the Degree of Restraint

During the filling of the model reservoir, in all the test series, no displacement occurred until the water level approached the crest of the model dam. The dam failed at this water level in the single monolith (SM) and low-restraint cases (FD and SK), with small or no pre-failure displacement. In the series with a moderate and high degree of restraint, however, a considerably higher water level was reached before the breach occurred, due to the support from the boundaries and interlocking between the monoliths. For these series, displacement commenced at a water level close to the crest level. Initially, the central monolith displaced straight downstream. The side monoliths did, however, rotate due to the boundary condition. The rotation led to pressure and increased friction in the joints, since the continued sideways displacement was retained due to the friction at the supports. The breach occurred suddenly in all cases. The dam started to slide and released a wave of water, which pushed the monoliths downstream. In many of the test series, the monoliths were pushed to the edge of the model area and collided with the steel bar used to protect the chute.

The failure loads of all the tests are presented in Figure 8, together with the averages and the sliding safety levels according to Equation (1). In the figure, the results are grouped into the individual tests with the test number presented. The failure load is close the crest of the dam (1200 mm) for the cases with low restraint but increases for the cases with moderate and high restraint. The load-bearing capacities in sliding were estimated using two friction coefficients: the assumed initial value ($\mu = 0.63$) and that derived from the wet winch test ($\mu = 0.56$); see Figure 6. With $\mu = 0.56$, Equation (1) gave a failure load of 1115 mm, which is a good prediction of the sliding stability of the single monolith (SM) ($p = 0.56$). This outcome is expected, since the SM test series was the final series, and the model had by then undergone significant wear. The initial friction value of $\mu = 0.63$ gave a failure load of 1200 mm, which is a good prediction for the sliding stability of the dam with moderate restraint ($p = 0.83$).

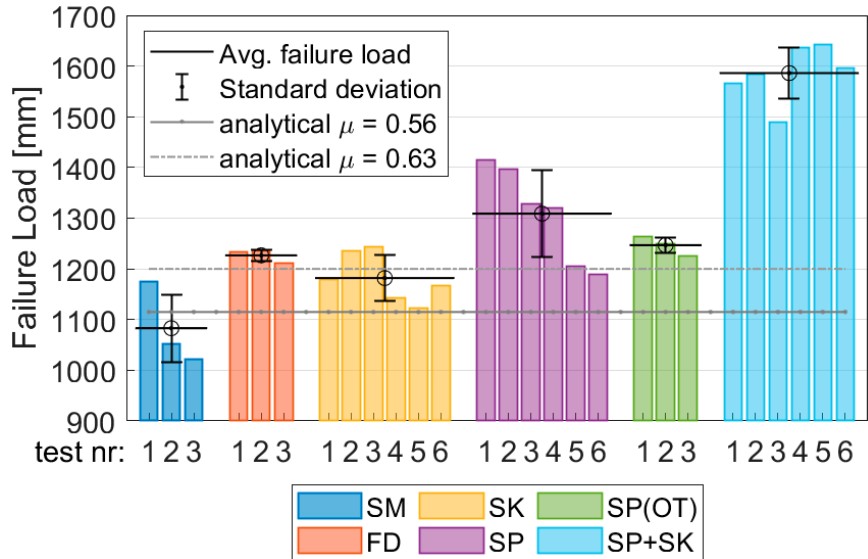

**Figure 8.** Failure loads for all tests, average failure loads, and standard deviation.

The safety factor, according to Equation (3), is presented in Figure 9a. The safety factor $(1 + \lambda)$ was normalized with respect to the average failure load of the single monolith (SM) case. The analytical load bearing capacity, according to Equation (1), was also normalized by the average failure load of the SM case. Compared to the single monolith case, the average failure load increased by 12% for the full dam failure (average of FD and SK), but the difference was not statistically significant ($p = 0.12$). With a moderate degree of restraint, the load-bearing capacity increased by 19% (average of SP and SP(OT)), while a high degree of restraint led to a 46% increase (SP+SK). Both increases were statistically significant ($p = 0.025$ and $p = 0.002$, respectively), emphasizing that lateral restraint markedly influences the load-bearing capacity.

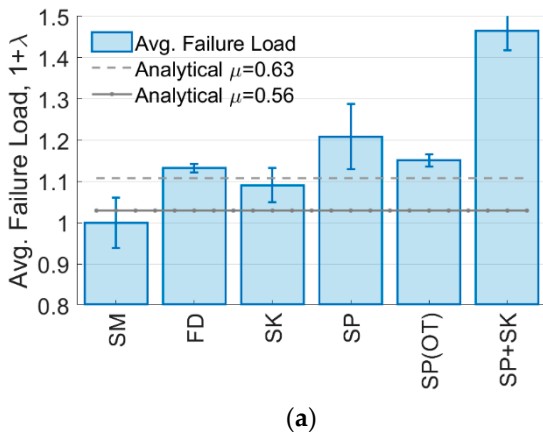
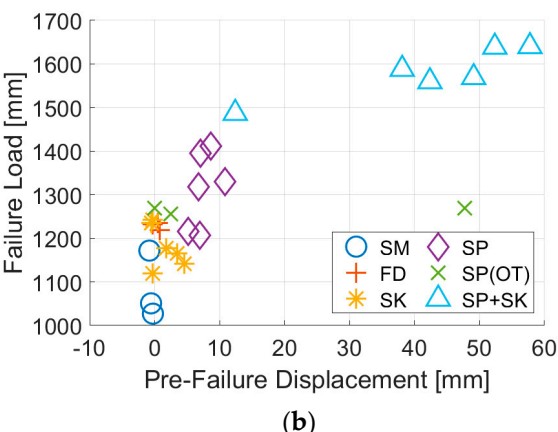

(**a**)    (**b**)

**Figure 9.** (**a**) Average failure load expressed according to Equation (3), normalized with respect to the average failure load in the SM case. The standard variation is indicated with error bars. (**b**) Failure load versus displacement at the point of failure.

Figure 9b presents the failure load versus pre-failure displacement for all the tests. The point of failure is defined as the time 0.3 s before the velocity of the model first reaches 100 mm/s. This gives a good representation of the initiation of the breach for all the tests. The lateral restraint clearly has a large impact on both the pre-failure behavior and the load-bearing capacity of the dam. Stiffer boundary conditions and a greater degree of interaction between the monoliths increase the pre-failure displacement. This is beneficial, as it raises the likelihood of an early warning. The pre-failure displacement was 50 mm for the SP(OT) 1 test. This is an outlier. The other tests in this series had pre-failure displacements of 5 and 7 mm. The reason for this outlier is difficult to explain, but it is probably due to monolith placement variations or random fluctuations.

### 4.2. Comparison of Failure Modes

In the single monolith (SM) test series, the central monolith was loaded to failure. The final position is shown in Figures 10a and 11a. The central monolith was physically separated from the adjacent monoliths by a distance of 20 mm and by using shear keys and side supports for the rest of the monoliths. Without physical separation, it was not possible to achieve the failure of a single monolith. This shows that the friction between the monoliths leads to a significant degree of interaction even without shear keys.

In the test series with low restraint, no supports were used at the boundary of the dam. The test was performed both with (SK) and without (FD) shear keys. In all the tests in this series, all the monoliths failed simultaneously, as shown in Figures 10b and 11b. This meant that the chute was blocked, and the water was not discharged. Instead, the water level remained roughly at the crest level of the monoliths and continued to exert pressure after the breach. The monoliths were displaced to the edge of the model area, colliding with the steel bar.

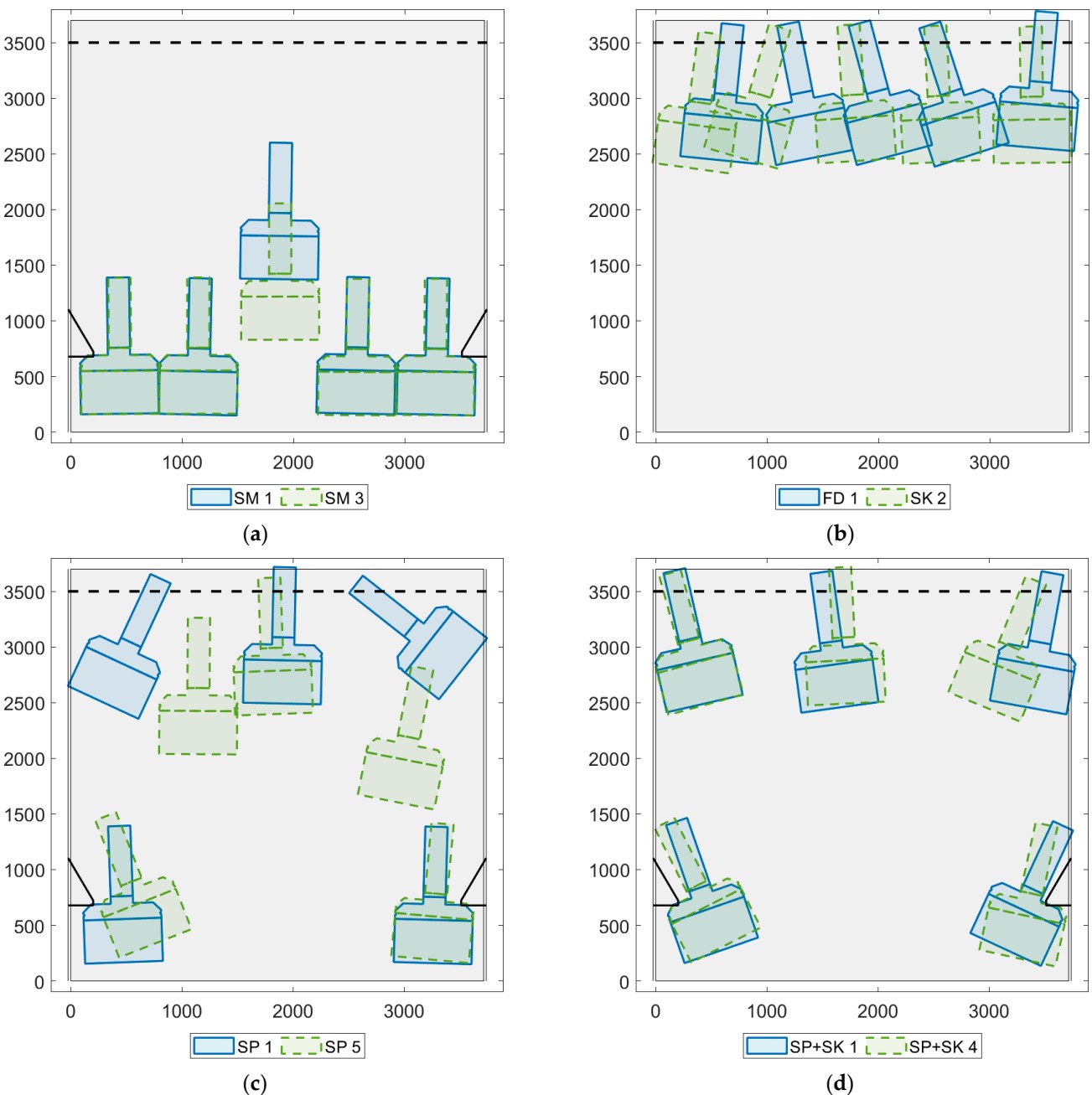

**Figure 10.** Top view of the final position of the monoliths for (**a**) a single monolith (SM), (**b**) a low degree of lateral restraint (FD and SK), (**c**) a moderate degree of lateral restraint (SP), and (**d**) a high degree of lateral restraint (SP+SK). The x- and y-axes are the distances from the lower left corner of the foundation slab in mm, and the dashed line is the protection rail. The individual test number is specified to highlight which test in the series is presented.

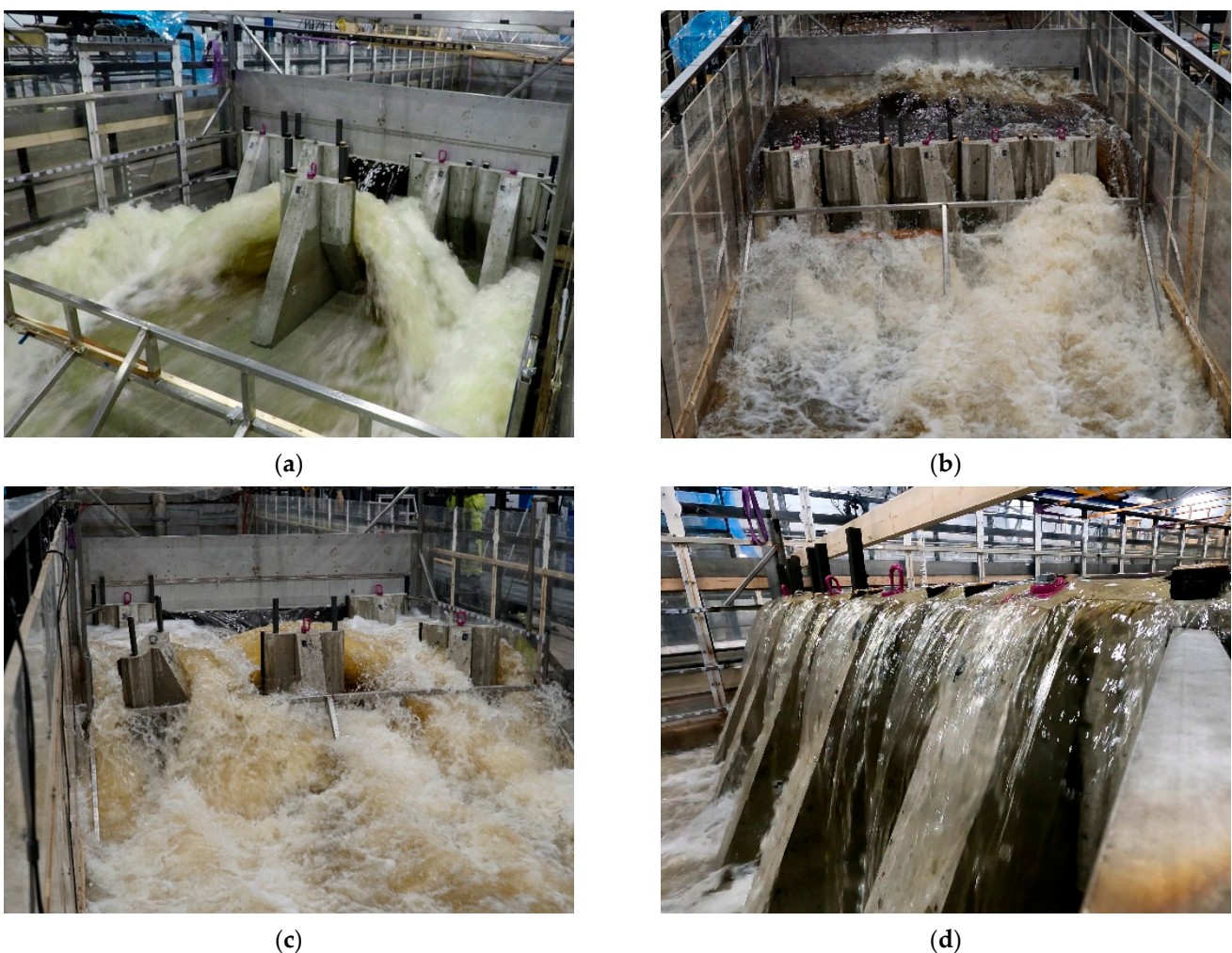

**Figure 11.** Illustration of failure modes: (**a**) single monolith (SM 1), (**b**) low degree of lateral restraint (FD 1), (**c**) moderate degree of lateral restraint (SP 1), and (**d**) overtopping in the SP(OT) 2 test. The individual test number is specified to highlight which test in the series is presented.

In the series of tests with a moderate degree of restraint (SP and SP(OT)), side supports were used but no shear keys. In the series with high restraint (SP+SK), both side supports and shear keys were used. In these test series, the three central monoliths were pushed to the edge of the model area and stopped by the steel bar, as shown in Figure 10c,d and Figure 11c. The side monoliths remained in position due to the supports but rotated. The failure occurred at a higher water level than in the previously presented test series and with a greater pre-failure displacement. The added restraint from the shear keys in SP+SK held the dam together and led to a large pre-failure displacement. The breach did not occur until the shear key in one of the joints failed. Therefore, the failure load depended on the strength of the ABS-plastic shear key. Two tests in this series are not presented since they were performed before the issue of the extension of the right-hand support was detected.

To simulate an overtopping failure, the watertight wall above the model dam was removed in the SP(OT) test series, as shown in Figure 11d. The model with a moderate degree of lateral restraint (SP) was used. The failure mode was similar to that in the SP series, and the difference in load-bearing capacity between the SP and SP(OT) series was not statistically significant ($p = 0.17$). The reservoir water level had to be raised more quickly in this test series to achieve and maintain the overtopping. The rate of increase in the water level was the same in SP 4 and 6 and the SP(OT) series. It was concluded that the rate of increase did not interfere with the load-bearing capacity (*t*-test: $p = 0.41 > 0.05$) or the failure mode. The load-bearing capacity and failure mode were similar between the

SP and SP(OT) test series indicating that the watertight wall above the model dam did not interfere with the results.

The discharge is related to the final position of the monoliths; see Figure 11. In the case of a single monolith failure (SM), the discharge was slow because the breach was small and the failed monolith blocked the outflow. In the test series with low restraint, all the monoliths failed simultaneously. After the monoliths collided with the steel bar, the dam blocked the chute with only small openings so that the reservoir drained slowly. In the test series with a moderate (SP) and high (SP+SK) degree of restraint, the breach was large, and the monoliths spread out over a large area. The discharge was, therefore, rapid, and the model reservoir drained quickly.

The average displacements and velocities during failures in the different test series are compared in Figure 12. The velocity is obtained via derivation of the displacement histories extracted with DIC. The displacement at the central point of the crest of Monolith 3 was used for comparison. An increased restraint increases both the hydrostatic pressure and the failure velocity. Compared to the single monolith case (SM), the low-restraint case (FD) had a 2.6 times higher velocity, the moderate restraint (SP) a 3.2 times higher velocity, and the high restraint (SP+SK) a 5 times higher velocity.

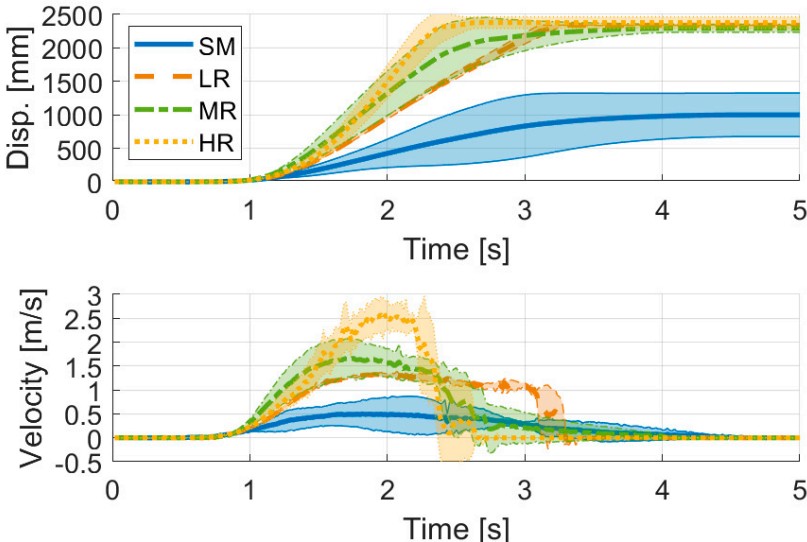

**Figure 12.** Average displacements and velocities during failures in the respective test series. The colored area shows the standard deviation ($\mu \pm \sigma$). SM: single monolith, LR: low restraint (average of FD and SK), MR: moderate restraint (average of SP and SP(OT)), and HR: high restraint (SP+SK).

### 4.3. Impact from Wear

A total of 42 tests were performed over a period of 5 weeks. This led to wear on the model, as shown by the fact that the friction varied throughout the testing period. Figure 13a shows the failure load for all the tests with a low degree of restraint in chronological order. The failure load increased during the first few tests and then decreased in the rest of the test series. This suggests that the friction in the dam–foundation interface varied between the test series, as this friction is the primary factor determining the failure load.

Visible changes were also noticed as the concrete surfaces of the dam–foundation interface became rougher during the initial tests. After several more tests, the surfaces became smoother with no detectable protruding grains. The friction test results shown in Figure 6 show that the dry friction angle decreased by 10% from an initial value of 33.5° to 30.5° after all the tests had been carried out. According to the analytical calculation (Equation (1)), this means a reduction in load-bearing capacity from 1200 mm to 1115 mm.

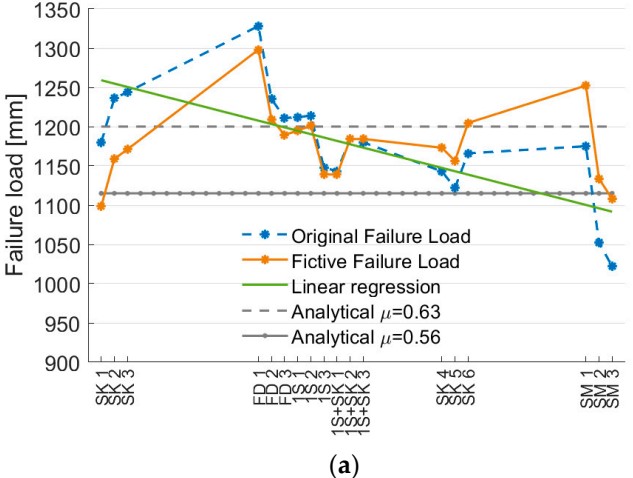
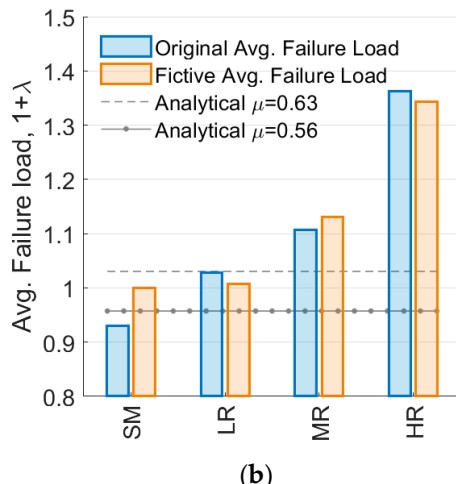

(**a**)                                  (**b**)

**Figure 13.** (**a**) Assumed linear relationship between the friction in the dam–foundation interface, and (**b**) average failure loads for all tests. The tests denoted 1S and 1S+SK use a single support (see Section 3.2). SM: single monolith, LR: low restraint (average of FD and SK), MR: moderate restraint (average of SP and SP(OT)), and HR: high restraint (SP+SK). The individual test number is specified to highlight which test in the series is presented.

The failure loads in all the tests with a low degree of restraint were normalized to take wear into account. A linear regression analysis was carried out, with the test numbers in chronological order being used as the *x*-axis. Figure 13a shows the normalized failure load for the tests using a low degree of restraint. Tests with a single support (1S and 1S+SK) were included to increase the sample size. These tests gave a failure behavior similar to the models using a low degree of restraint (FD and SK) but are not presented in detail in this article (see Section 3.2). Figure 13b shows both the average failure load for all the test series and the adjusted averages. The failure loads $(1 + \lambda)$ were normalized with respect to the adjusted SM test series. The adjusted averages suggest a comparable level of safety for the single monolith failure (SM) and the failure of the test series with low restraint (FD and SK). The figure shows that, with the original friction value of $\mu = 0.63$, Equation (1) gave a good estimate of the load-bearing capacity for the single monolith (SM) and low restraint cases (FD and SK). With the adjusted averages, the load-bearing capacity increased by 13% in the case with moderate restraint (SP and SP(OT)) and 34% in the case with high restraint.

## 5. Discussion

Although the design of the model was idealized, the geometry and boundary conditions will greatly affect the behavior during a specific dam failure. The results do, however, show the general 3D behavior of concrete buttress dams. The boundary conditions and the interaction between the monoliths provide lateral pressure and thereby determine the inter-monolithic interaction in the joints through friction. The transfer of shear force between the monoliths due to friction was large, but the shear keys contributed to an even higher shear transfer. Without shear keys, the load was primarily transferred through friction in the physical model tests, but some mechanical interlocking also occurred, since the joint surfaces had some roughness. The lateral restraint and interaction between the monoliths enhanced the load-bearing capacity of the dam and influenced its pre-failure behavior. A greater increased interaction between the monoliths could therefore be beneficial for the structural safety of the dam structure, although this may increase the breach size during failure.

In the model tests, the water level in the reservoir was increased to drive the model dam to failure. In most of the tests, failure occurred at a reasonable water level near the crest of the dam, but a higher failure load was observed when the lateral restraint was increased. The maximum average overtopping was 32%, a value that is unreasonable for a

dam of this size. The failure mode can, therefore, be questioned, but it indicates the added strength due to the 3D effects.

The failure of a real dam could be caused by a rapidly increasing water level, such as a flood wave caused by the collapse of an upstream dam, heavy rain in a small reservoir, or a landslide. Another likely failure scenario would be caused by exceptional ice loads, a loss of friction or structural strength due to leaching in cracks or joints, material transport in rock joints, or erosion during an overtopping scenario. These effects can affect different parts of the dam or individual monoliths differently and make the prediction of real dam failure more difficult than these model tests suggest.

The model considered rigid body motions, and the material of the model dam was not scaled as a result. However, the shear keys, which were produced from ABS plastic, broke during the tests using a high degree of restraint, which suggests a risk of concrete crushing at the joints, which could lower the load-bearing capacity. Further investigations should be conducted on the stress levels of the shear keys using FE analyses, but this was outside of the scope of this study. The effect of cohesion, real friction levels, the roughness of the rock surface, and fracture planes in the concrete and rock should also be further explored.

In the tests using a low degree of lateral restraint, the dam failed suddenly without any early indications of displacement. When the degree of lateral restraint was increased, the pre-failure displacement increased. The breach was, however, still quite sudden. The rock in the model was idealized, and the foundation of a real dam will be rough. In addition, many Swedish buttress dams have reinforcement in the form of rock bolts or prestressed tendons. In this case, failures will be more ductile than a pure frictional failure along a flat surface. There would thus be a greater likelihood of receiving an early warning on a real dam in the event of the initiation of failure. However, these early warnings require automatic and continuous monitoring of the dam structure to measure global behavior. They also require threshold values and alarm limits to be defined. Although the current study does not include a method for applying alarm limits, it is hoped that the results of these model tests will aid in establishing such limits in the future.

## 6. Conclusions

The load-bearing capacity in the single monolith case and in the cases without side supports was predicted well with the analytical sliding stability criterion, but when the side supports were introduced and the lateral restraint was increased, the analytical expression was no longer applicable. The results showed that the interaction between the monoliths greatly affected the failure mode, load-bearing capacity, and pre-failure displacements. The breach size and the load-bearing capacity of the dam depend on the ability to transfer shear forces between the monoliths. The transfer of shear forces depends on the contact pressure in the vertical joint between the monoliths and the presence of shear keys. The contact pressure appears when the monoliths start rotating at the onset of failure and is maintained by the boundary conditions. Therefore, 3D effects, such as the dam length, distance to the boundaries, and type of connections at the boundaries, should be taken into consideration in the stability analysis. This applies both in the design and evaluation of existing dams.

In the tests with a low degree of lateral restraint, the failure was sudden and occurred without warning or initial deformation. This makes a potential failure more difficult to detect using traditional dam instrumentation and results in a sudden appearance of a flood wave. However, rock bolts were not used, and the rock surface was simplified and smooth. For more complex geometries, the pre-failure deflections might be larger, and the time of failure might be longer. The dataset resulting from the physical model tests can potentially aid in the future development of dam monitoring alarm limits and threshold values.

In the physical model tests, the shear transfer between the monoliths was significant. It was difficult to initiate failure in a single monolith, even without shear keys in the expansion joints. Failure could only occur in a single monolith when it was physically separated from the rest of the dam. This indicates that a failure in a buttress dam is likely to encompass several monoliths, and the assumption during a flooding analysis that a

single monolith fails is unrealistic. The physical model tests represented an idealized case and cannot be used for the prediction of a potential failure in any specific dam. Methods, preferably numerical, for predicting the failure geometry of concrete buttress and gravity dams should therefore be further examined.

**Author Contributions:** Conceptualization, R.M., E.N. and J.E.; Methodology, R.M., E.N., A.S. and J.E.; Software, J.E.; Validation, J.E.; Formal Analysis, J.E.; Investigation, J.E.; Resources, E.N.; Data Curation, J.E.; Writing—Original Draft Preparation, J.E.; Writing—Review and Editing, R.M., E.N., A.S. and A.A.; Visualization, J.E.; Supervision, R.M., E.N., A.S. and A.A.; Project Administration, R.M. and A.A.; Funding Acquisition, R.M., E.N. and A.A. All authors have read and agreed to the published version of the manuscript.

**Funding:** The research presented in this thesis was carried out as a part of "SVC—Swedish Centre for Sustainable Hydropower". SVC has been established by the Swedish Energy Agency, Energiforsk, and Svenska Kraftnät together with Luleå University of Technology, KTH Royal Institute of Technology, Chalmers University of Technology, Uppsala University the Faculty of Engineering at Lund University, Karlstad University, and Umeå University. Participating companies and associations are AFRY, Andritz Hydro, Boliden, Fortum, Holmen Energi, Jämtkraft, Jönköping Energi, Karlstads Energi, LKAB, Mälarenergi, Norconsult, Rainpower, Skellefteå Kraft, Statkraft, Sweco Sweden, Tekniska verken i Linköping, The Hydroelectric Environmental Fund, Uniper, Vattenfall R&D, Vattenfall Vattenkraft, Voith Hydro, WSP Sverige, and Zinkgruvan.

**Data Availability Statement:** The data presented in this study are available on request from the corresponding author.

**Conflicts of Interest:** The authors declare no conflict of interest.

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
