# Peer review of "Physical Model Tests of Concrete Buttress Dams with Failure Imposed by Hydrostatic Water Pressure"

_water, doi:10.3390/w15203627_

Round 1

Reviewer 1 Report

General comments:

This paper presents a study carried out with the main objective of contributing to a better understanding of the mechanism of failure due to sliding along the dam/foundation interface. The paper is very interesting. Although the study has some limitations, which are adequately pointed out, the tests provided valuable information and several conclusions are drawn which can be effectively used in practice. The paper is very well written and organized. Only a few remarks are pointed out.

Specific comments:

  1. Section 3.2, page 7, line 6 of the first paragraph: the sentence “the case without shear keys and side supports” could be replaced by “the case without shear keys and without side supports”.
  2. Section 3.2, page 7, at the end of the second paragraph: the sentence “Rock and soil erosion behind the dam toe …” could be replaced by “Rock and soil erosion downstream from the dam toe…”
  3. In Figure 9 and in Figure 10:  I believe that the number after the acronym of each test is the number of the test in each series (varies from 1 to 3 or from 1 to 6). This should be clear.
  4. Section 4.3. The authors mention that 42 tests were carried out. However, only 27 tests are presented in both Table 2 and Figure 7. The results of the remaining tests were disregarded or were also analysed­?
  5. Figure 11: In order to make this figure clear the abbreviations LR, MR and HR should be included in Table 2, or explained, as in Figure 12.
  6. Figure 12a : I believe that tests 1S and 1S+SK are tests carried out using a single support. This has to be explained.
  7. Section 4.3: In the third line of the first paragraph and in the 5th line of the third paragraph the authors mention that “Figure 12a compares … with a low degree of restraint”. However, results for moderate restraint and high restraint are also shown. Please revise both sentences.
  8. Section 5, 3rd paragraph: the authors mention that “Another likely failure scenario would be caused by increased uplift pressure, …”. However, in buttress dams the spaces between buttresses allow the discharge of water seeping through the foundation, thus greatly reducing uplift pressures. Uplift pressures may be a problem in gravity dams but are unlikely to occur in buttress dams.
  9. Section 5, last paragraph: as far as I know, most of the dams do not have reinforcement in the form of rock bolts or prestressed tendons.
  10. References:

10.1     Reference 12: Please check the order of authors.

10.2     Reference 13: Faria, instead of Fariab.

10.3     Reference 15: Please check the title of the document.

Author Response

Thank you for the thorough review. Please see the attached PDF file with the response to your comments. 

Reviewer 2 Report

I thoroughly read the article file. There are drawbacks that I will address in the comments below.

# Abstract:

·         This abstract is not scientific and focuses on generalities. The abstract should be an extract from your research so that any audience can understand it by reading it.

·         You have even acted poorly in expressing the results. At the beginning of the abstract, a brief introduction should be given and then the existing problems. Then briefly describe your goals and then bring your materials and methods, and results.

·          In this section, the only general content is provided and no attention is paid to details.

·         Add the important and key results to the abstract. Also, bring significant results to your work so that the audience has a clear understanding of the outputs of this article. All these points must be applied.

# Introduction:

·         Expressing the novelty of the article and the difference with the work of others should be well expressed.

·         You need to add what your article will solve in the field at the end of the introduction.

·         Please cite below mentioned paper in the importance of studying hydraulic structures. https://doi.org/10.3390/w15020314.

# Materials and Methods

·         What is the work to get the accuracy of measurement and avoid the scale effect? It should be added paragraph to explain the impact of scale effect.

·         The boundary conditions are not well described. The way of presenting the boundary conditions in the following research should be investigated  and cited. https://doi.org/10.1007/s41062-023-01083-z

# Results:

·         You have concluded very quickly and no analysis has been done on the outputs. Your results section is incomplete.

·         The engineering application of the results obtained by this manuscript should be highlighted at the end of the results.

·         The description of figures 7 and 8 are not complete. Authors should add sufficient explanations.

·         Please explain the behavior of shear forces between the monoliths.

·         https://doi.org/10.1007/s41062-023-01083-z

 I thoroughly read the article file. There are drawbacks that I will address in the comments below.

# Abstract:

·         This abstract is not scientific and focuses on generalities. The abstract should be an extract from your research so that any audience can understand it by reading it.

·         You have even acted poorly in expressing the results. At the beginning of the abstract, a brief introduction should be given and then the existing problems. Then briefly describe your goals and then bring your materials and methods, and results.

·          In this section, the only general content is provided and no attention is paid to details.

·         Add the important and key results to the abstract. Also, bring significant results to your work so that the audience has a clear understanding of the outputs of this article. All these points must be applied.

# Introduction:

·         Expressing the novelty of the article and the difference with the work of others should be well expressed.

·         You need to add what your article will solve in the field at the end of the introduction.

·         Please cite below mentioned paper in the importance of studying hydraulic structures. https://doi.org/10.3390/w15020314.

# Materials and Methods

·         What is the work to get the accuracy of measurement and avoid the scale effect? It should be added paragraph to explain the impact of scale effect.

·         The boundary conditions are not well described. The way of presenting the boundary conditions in the following research should be investigated  and cited. https://doi.org/10.1007/s41062-023-01083-z

# Results:

·         You have concluded very quickly and no analysis has been done on the outputs. Your results section is incomplete.

·         The engineering application of the results obtained by this manuscript should be highlighted at the end of the results.

·         The description of figures 7 and 8 are not complete. Authors should add sufficient explanations.

·         Please explain the behavior of shear forces between the monoliths.

·         https://doi.org/10.1007/s41062-023-01083-z

Author Response

(The authors gave the same response as above.)

Reviewer 3 Report

This article uses a large-scale experimental setup to simulate dam failure. Experiments at this scale are difficult to design and perform, this in itself makes this work original, the results of these experiments are interesting and can be used for the safe design of similar structures. I would suggest in a future work to compare the results of the experiment with some real case of dam failure for backward failure analysis. Some further suggestions are:

·        In second paragraph of introduction: …analytical stability analysis methods [3,4].

·        In the first paragraph of page 2: … the breach size and the breach development time…(also, correct in the last paragraph)

·        In page 3 before eq. (1): …sliding stability safety factor…

·        In table 1 scale factors: The time scale factor is wrong. Were they have been used?

·        In figure 1a: the mm unit is missing.

·        In figure 8b the data shows a nice exponential relationship.

·        In figure 9 caption: Top view of final position…

·        The paper is missing a photo showing some real dams that this experiment is designed to simulate.

Author Response

(The authors gave the same response as above.)

Reviewer 4 Report

In this investigation, Physical model tests were conducted to enhance understanding of the failure behavior of concrete dams and to help improve methods for stability assessments and failure prediction. A model buttress dam, scaled 1:15, consisting of 5 monoliths measuring 1.2 m in height and 4 m in width, was developed and loaded to failure using water pressure. The model dam had detachable abutment supports and shear keys to allow for variation in the 3D behavior. The results showed that the failure of a single dam monolith was unlikely and that a higher degree of lateral restraint gave both a higher failure load and a better indication of an impending failure. These findings suggest that the entire dam, including realistic boundary conditions, should be considered during stability assessment. The results also suggest that assuming the failure of a single monolith during flooding analysis, which is a common assumption in the dam safety codes, is unconservative.  In this regard, this study will be beneficial to the literature. It can be accepted after minor changes.

1.      Language should be reviewed in general. Some places are difficult to understand by the reader.

2.      Introduction should be expanded. Especially, it should be added more references such as

3.      What is the novelty of this study? It's not exactly obvious. Novelty should be given at the end of Section 1.

4.      Authors said that “Digital Image Correlation (DIC) was employed to recreate the displacement at any given point along the model dam. The displacements from the DIC were calibrated with the results from the wire gauges. ” What is the main ıdea to choose this? It should be given more information about the DIC and given more pictures in the test setup ?

5.      Figure 11  is not clear. It should be given more clear.

6.      The results should be discussed in detail. Presentations should not be made in the form of a technical report.

7.      Conclusion should be given more specific results and discussions. It is so long.

8.      References should be updated. Especially authors should add the extra new researches.

Moderate editing of English language required

Author Response

(The authors gave the same response as above.)

Round 2

Reviewer 1 Report

I think that the additional information provided by the authors contributes to the clarity of the manuscript and that this paper can be accepted in its present form.

Reviewer 2 Report

The authors have provided the reviewer's answer well. Also, they have provided general changes in the manuscript.

At the discretion of the respected editor, the manuscript can be accepted.

Thanks

Reviewer 3 Report

No further comments, I aggree with the modifications made.

Reviewer 4 Report

Every modification was made. It is acceptable.